# Monoclonal Gammopathies of Clinical Significance: A Critical Appraisal

**DOI:** 10.3390/cancers14215247

**Published:** 2022-10-26

**Authors:** Rafael Ríos-Tamayo, Bruno Paiva, Juan José Lahuerta, Joaquín Martínez López, Rafael F. Duarte

**Affiliations:** 1Hospital Universitario Puerta de Hierro, Fundación para la Investigación Biomédica del Hospital Universitario Puerta de Hierro-Majadahonda, 28222 Majadahonda, Spain; 2Centro de Investigación Médica Aplicada, Instituto de Investigación Sanitaria de Navarra, Clínica Universidad de Navarra, 31008 Pamplona, Spain; 3Hospital Universitario 12 de Octubre, Instituto de Investigación del Hospital Universitario 12 de Octubre, 28041 Madrid, Spain

**Keywords:** monoclonal gammopathy of clinical significance, diagnosis, prognosis, treatment, amyloidosis AL

## Abstract

**Simple Summary:**

Monoclonal gammopathy of clinical significance (MGCS) refers to a recently coined term describing a complex and heterogeneous group of nonmalignant monoclonal gammopathies. These patients are characterized by the presence of a commonly small clone and the occurrence of symptoms that may be associated with the clone or with the monoclonal protein through diverse mechanisms. This is an evolving, challenging, and rapidly changing field. Patients are classified according to the key organ or system involved, with kidneys, skin, nerves, and eyes being the most frequently affected. However, multiorgan involvement may be the most relevant clinical feature at the presentation or during the course. This review delves into the definition, history, differential diagnosis, classification, prognosis, and treatment of this group of entities by analyzing the evidence accumulated to date from a critical perspective.

**Abstract:**

Monoclonal gammopathies of clinical significance (MGCSs) represent a group of diseases featuring the association of a nonmalignant B cells or plasma cells clone, the production of an M-protein, and singularly, the existence of organ damage. They present a current framework that is difficult to approach from a practical clinical perspective. Several points should be addressed in order to move further toward a better understanding. Overall, these entities are only partially included in the international classifications of diseases. Its definition and classification remain ambiguous. Remarkably, its real incidence is unknown, provided that a diagnostic biopsy is mandatory in most cases. In fact, amyloidosis AL is the final diagnosis in a large percentage of patients with renal significance. On the other hand, many of these young entities are syndromes that are based on a dynamic set of diagnostic criteria, challenging a timely diagnosis. Moreover, a specific risk score for progression is lacking. Despite the key role of the clinical laboratory in the diagnosis and prognosis of these patients, information about laboratory biomarkers is limited. Besides, the evidence accumulated for many of these entities is scarce. Hence, national and international registries are stimulated. In particular, IgM MGCS deserves special attention. Until now, therapy is far from being standardized, and it should be planned on a risk and patient-adapted basis. Finally, a comprehensive and coordinated multidisciplinary approach is needed, and specific clinical trials are encouraged.

## 1. Introduction

Monoclonal gammopathies (MGs) are a wide and heterogeneous group of conditions characterized by the presence of a monoclonal (M) protein in the peripheral blood and/or in the urine. The synthesis and release into the plasma of the M-protein, in most cases, is an expression of an underlying plasma cell (PC) neoplasm (PCN).

PCNs are clonal B-cell tumors that range from asymptomatic and stable disorders to diseases with extensive end-organ damage. This great heterogeneity translates to clinical decisions that range from periodic observation to the urgent initiation of anti-clonal therapy. The World Health Organization (WHO) classification of lymphoid tumors periodically provides a global reference for the diagnosis of lymphoid neoplasms. The fifth edition (WHO-HAEM5) has just been released [1]. Remarkably, its definitions have not only been adopted for clinical and research use, but they have also been incorporated into the International Classification of Diseases. Therefore, it correspondingly serves as a universal position for epidemiological analysis and monitoring across international health policy organizations on five continents. The updated WHO-HAEM5 classification of PCNs, now called PCNs, and other diseases with paraproteins is shown in Table 1.

On the other hand, the International Consensus Classification of Mature Lymphoid Neoplasms has also just been revised [2]. The definition, work-up recommended tests, and diagnostic criteria for these entities have been upgraded. Table 2 shows the International Consensus Classification of Mature B-cell Neoplasms.

MG of undetermined significance (MGUS) is the most frequent MG, representing about 55% of all MGs [3]. MGUS is a common finding in daily clinical practice. The prevalence of MGUS is about 3.2% in people 50 years of age or older [4], but it depends on age, race, family history, and the screening method used. Growing evidence shows that persons with MGUS have an increased risk of infection [5], thrombosis [6], and fractures [7,8]. Overall, people with MGUS have a global risk of progression of approximately 1% per year [9], with IgM and non-IgM different modes of progression [10]. A common risk score for estimating the probability of progression is based on the serological subtype of MGUS, the M protein level, and the free light chain (LC) ratio (FLCr) [11]. This well-known Mayo Clinic model was validated in a later Swedish study, confirming the predictive value of a high M-protein level and abnormal FLCr. This study also showed that the addition of classical immunoparesis to the previous set of prognostic variables increases the discriminatory power of the model, helping to identify high-risk (vs low-risk) MGUS patients [12]. Additional variables may be considered to reach a more precise prediction [13,14,15,16,17,18,19,20,21,22,23,24,25,26,27], although the role of some of them remains controversial. For this reason, patients must be followed in a risk-adapted approach, although follow-up recommendations are heterogeneous [28]. The most relevant variables that influence the risk of progression in MGUS are shown in Table 3.

Non-immunoglobulin M (IgM) MGUS is a virtually universal precursor to multiple myeloma (MM) [29,30], whereas every MGUS subtype can progress to light chain (AL) amyloidosis. As bone marrow examination may be the only factor that will upstage an MGUS to Smoldering MM, this must be strongly considered in the evaluation of all MGs.

Regarding AL amyloidosis, urinary albumin excretion and serum creatinine, with estimated glomerular filtration rate, can efficiently detect early kidney involvement, while N-terminal pro–brain natriuretic peptide has 100% sensitivity, although it is not specific, in detecting early, reversible, cardiac dysfunction caused by amyloidogenic LC [31].

Most individuals with MGUS are asymptomatic at the time of diagnosis. Nevertheless, the apparition of symptoms in the presence of proven organ damage may point out the existence of several conditions associated with clonal Ig secretion in the absence of overt malignancy, globally termed MG of clinical significance (MGCS). The group of MGCS most extensively studied until now is MG of renal significance (MGRS). The ongoing description of several groups of MGCS adds complexity to the management of patients with suspected newly diagnosed MG.

Controversy remains in key aspects related to the definition, classification, diagnosis, and therapy of MGCS patients. The aim of this work is to update and critically review the current evidence about the diagnosis, prognosis, and therapy of this group of entities. This increasingly complex scenario must be faced through a multidisciplinary and risk-adapted approach, a standardized workup, an evidence-based follow-up, and the use of consensus recommendations, aiming to achieve an early and accurate diagnosis and finally providing optimal clinical management to these patients.

## 2. Definitions

### 2.1. MGUS

MGUS is a premalignant, clonal PC disorder characterized by the presence of an M-protein produced by a small B-cell/PC clone in persons without features of symptomatic disease related to malignant disorders [32,33,34,35,36,37]. MGUS can be classified into 3 main subtypes, Ig M MGUS, non-IgM MGUS, and LC MGUS, based on a different pattern of progression.

#### 2.1.1. Non-Ig M MGUS

Non-Ig M MGUS is defined by a serum M-protein less than 3 g/dL and less than 10% of clonal bone marrow PC (cBMPC), and the absence of symptoms or end-organ damage attributed to the underlying lymphoplasmacytic disorder.

#### 2.1.2. Ig M MGUS

Ig M MGUS has the same definition as Non-Ig M MGUS, except for the type of bone marrow cells to assess. In this case, less than 10% of cBMPC and a lack of lymphoplasmacytic B-cell aggregates should be demonstrated. In accordance with recent international consensus recommendations, two subtypes of Ig M MGUS should be differentiated: IgM MGUS of PC type, which is considered a precursor of MM, and IgM MGUS, NOS, including all cases with a MYD88 mutation, those with detectable monotypic/monoclonal B-cells but without abnormal lymphoplasmacytic aggregates diagnostic of lymphoplasmacytic lymphoma, and those lacking evidence of other small B-cell neoplasms. Hence, MYD88 mutation should be incorporated into routine workups [2,38,39,40,41].

#### 2.1.3. LC MGUS

LC MGUS patients have an abnormal FLCr, <0.26 or >1.65, with increased level of the involved LC without Ig heavy chain expression on immunofixation, BMPC less than 10%, urinary M-protein less than 500 mg/24 h, and again, the absence of end-organ damage or amyloidosis [42,43].

### 2.2. MGRS

MGRS is basically defined as a kidney disease related to the presence of an M-protein, diagnosed by demonstration of monoclonal deposits in the kidney biopsy. Monoclonal deposits can consist of monoclonal LC, heavy chain, or intact Igs. Restriction to a single class of LC and/or heavy chain is mandatory [44,45,46,47,48,49].

### 2.3. MGCS

MGCS is a heterogeneous group of nonmalignant MGs featuring two main characteristics: the presence of a clone, which is commonly associated with the secretion of an M-protein, and symptoms that can be related to the M-protein or to the clone itself by different mechanisms that do not include tumor burden. The MGCSs are best divided into different systems that are affected, the most common of which are kidney, nerve, and skin. However, a systemic and multiorgan presentation is frequent at the time of diagnosis or during the course of the disease [50,51].

## 3. Historic Note

MGUS is a young concept. The term was coined less than a half century ago [52]. The first studies led by the Mayo Clinic group laid the foundations for an understanding of this entity in terms of epidemiology and clinical impact, but our knowledge about it has been limited until the current century. The first prospective population-based MGUS screening study was published in 2009, showing that MM is consistently preceded by the MGUS precursor stage [29]. MGUS is a heterogeneous disorder associated with a higher risk of other medical problems. Despite the availability of diagnostic criteria, there has been a notable lack of consensus in recent years on key aspects, such as performing a diagnostic bone marrow biopsy and aspirate [35], and this controversy has reached the present day.

MGRS was described for the first time in 2012 [44]. A subgroup of MGUS patients had a condition that was no longer undetermined or insignificant. The term MG of cutaneous or skin significance (MGSS) emerged four years later [53]. These concepts paved the way to the broader and unifying concept of MGCS, which was introduced in 2018 [50]. This term encompasses a wide range of entities, including patients who paradoxically present a small clone and a very low protein level but with significant organ damage that can be life-threatening. In 2019, the term MG of ocular significance (MGOS) was proposed [54]. A schematic timeline representation of the description of relevant MGCS disorders is shown in Figure 1.

## 4. A Practical Classification of MGCS

Despite the increasing knowledge and interest in MGs observed in recent years, MGRS were not included in the last International Classification of Diseases 11th Ed. 2021 or in the previous WHO-HAEM4, except for Monoclonal Ig deposition diseases. MGRS have been most recently included in the 2022 WHO-HAEM5 classification.

As a concept, the key difference between MGUS and MGCS is the presence of disease or organ damage associated with the M-protein or to the clone itself, with therapeutic implications in the last one. Therefore, individuals with MGUS are asymptomatic, without organ damage attributable to the clone or the M-protein, and without the need for treatment, whereas patients with MGCS are symptomatic, organ damage associated with the clone itself or the M-protein can be confirmed, and treatment is needed. Despite recent progress, the link between organ damage and the underlying clone or the M-protein is difficult to prove for some of these entities, for which the pathophysiological mechanism is poorly understood or unknown.

From a practical point of view, MGCS can be divided according to the main organs involved: MGRS, MGSS, MGOS, and MG of neurological significance (MGNS). It remains to be demonstrated whether other groups of MGCS will be unveiled in the coming years.

## 5. Monoclonal Gammopathies of Renal Significance

This group of disorders is now included in the subgroup of MGs of the PCN and other diseases with paraproteins (WHO-HAEM5), besides IgM MGUS, non-IgM MGUS, and cold agglutinin disease [1].

Kidney biopsy plays a central role in this diagnosis [55,56,57]. Strikingly, in a study of 6300 patients with MG, only 160 (2.5%) had undergone a kidney biopsy. Of the 160 patients, 64 (40%) had an MGRS lesion, with AL amyloidosis the most common finding, accounting for 43.8% of these lesions. In this study, the decision to perform a kidney biopsy depended on age, the level of proteinuria, and renal function. Despite some limitations, this study showed that the likelihood of performing a kidney biopsy was higher in younger patients with more severe kidney disease in terms of proteinuria and serum creatinine. The probability of reaching a diagnosis of MGRS was associated with the presence of significant proteinuria (≥1.5 g/day), hematuria and an abnormal FLCr. Therefore, a kidney biopsy should be particularly considered in patients with these characteristics [58]. A kidney biopsy is usually indicated for significant proteinuria and/or renal insufficiency of unknown origin, but the presence of a MG should increase the likelihood of the indication, provided this group of diseases is diagnosed by demonstration of monoclonal deposits in the kidney. MGRS is a dynamic concept, and the classical classification of MGRS-associated renal lesions has undergone an evolution in recent years [45,48]. The current pathologic classification of MGRS-associated diseases is shown in Table 4.

The list of MGRS-associated kidney disorders is still expanding. These conditions can manifest as glomerular diseases, tubulopathies, and vascular involvement, with varying clinical presentations. Therefore, diagnosis is often challenging due to the heterogeneous presentation, the difficulty in establishing a pathogenic link between the presence of the M-protein or serum FLC and kidney disease, and the decision to perform a kidney biopsy. The high incidence of MGUS and other kidney disorders in elderly patients makes the diagnostic process even more difficult. Treatment can potentially reverse kidney disease; hence, early diagnosis is of great value [59,60]. A combined hematologic and nephrologic approach is crucial to establishing the causative role of the M-protein in the pathogenesis of kidney disease. Regarding the risk of progression to MM, MGRS had a significantly higher risk than MGUS (18% vs. 3%; *p* < 0.001), and this risk was 10% vs. 1% within the first year after diagnosis [61].

Several studies have shown that the expected evolution of most MGRS patients is characterized by the demonstration of progressive renal failure until its final stage. The timely administration of anti-clonal therapy may help to avoid this inexorable course in some cases. The goal of therapy should focus on preventing further renal damage by the M-protein and reaching a hematologic response because hematologic responses are a prerequisite to achieving renal responses [62,63,64,65,66,67].

The largest study on biopsy-proven MGRS has recently been reported. This is a retrospective international real-world analysis of 280 patients, of which 180 were AL amyloidosis-related (MGRS-A) and 100 were non-amyloidosis (MGRS-NA). The most frequent subtype in MGRS-NA was M-Ig deposition (53%). The clinical behavior of the MGRS-A group was poorer than that of the other groups. This was probably due to the huge prognostic impact of cardiac involvement in AL amyloidosis patients, despite a lower cBMPC infiltration and better renal function. The lack of or delay in treatment also had a negative prognostic impact. The study highlights the importance of hematologists and nephrologists working together. However, one out of four patients in this study had no hematologic and/or renal evaluation of response. Therefore, there is room for improvement in the management of MGRS patients [68].

## 6. Monoclonal Gammopathies of Skin Significance

Some dermatologic entities are strongly associated with the presence of a MG, and they should be referred to as MGSS. Again, the demonstration of the association between the M-protein or the clone itself with skin damage is key. As expected, a skin biopsy plays a critical role in the diagnostic process. The direct toxicity of M-protein, host immune abnormalities, specific cytokines, and PC infiltration can, among other mechanisms, produce severe skin manifestations. Thus, MGSS has been classically divided into four different groups according to the type of clone and the type of association between the cutaneous disorder and the underlying MG [69,70] (Table 5):

The presence of unexplained skin lesions in a patient with MG should be assessed primarily by a hematologist and a dermatologist, with the participation of other specialties, such as rheumatology, internal medicine, and ophthalmology, in certain situations. Every patient with MG and a new skin lesion of unknown origin should be investigated. In these cases, a skin biopsy and bone marrow examination, in addition to the standard laboratory evaluation of PC dyscrasia, should be performed [70]. Only two of these disorders will be described to emphasize the complex clinical presentation as a syndrome and the need to fulfil the diagnostic criteria.

POEMS is a rare paraneoplastic syndrome associated with a PCN [1]. Its acronym stands for polyradiculoneuropathy (P), organomegaly (O), endocrinopathy (E), monoclonal gammopathy (M), and skin changes (S). The components of this pentad are not always required for diagnosis. This entity is also called osteosclerotic myeloma due to the presence of osteosclerotic bone lesions in most patients. Other features may include lymphadenopathy, and in these patients, a PC variant of Castleman disease may be present. Papilledema, pleural effusion, ascites, erythrocytosis, and thrombocytosis can also be detected. POEMS syndrome is defined by a set of diagnostic criteria (Table 6), of which both major criteria must be present, together with another major (other) criterion and one minor criterion [71,72]. Most POEMS patients had lambda LC. The skin lesions associated with POEMS syndrome occur in more than two-thirds of patients, including most commonly hyperpigmentation and hypertrichosis, although acrocyanosis, plethora, telangiectasia, and skin thickening have also been reported. Rarer skin lesions may also be present in some patients. Serum vascular endothelial growth factor is elevated in most patients, being the levels correlated with disease activity, and in some studies, a causative relationship with skin lesions has been suggested.

Schnitzler syndrome is a rare entity described in 1972 and characterized by chronic urticaria, which may be non-pruritic, associated typically with an IgM, mainly kappa, MG. The diagnosis is made based on clinical features. A skin biopsy usually shows neutrophil urticarial dermatosis without signs of vasculitis. The diagnosis of this entity is established when the two mandatory major criteria and at least two (IgM cases) or three (in IgG cases) minor criteria of the so-called Strasbourg criteria are present (Table 7) [73,74]. Only around 300 patients have been reported to date, but it is probably underdiagnosed. The prognosis of MG in Schnitzler syndrome is generally favorable, although the quality of life is generally poor due to associated morbidity related to rash, fever, and pain. Lymphoplasmacytic lymphoma develops in approximately 15% of patients; thus, close follow-up of these patients is warranted. Although anti-interleukin-1 therapy is now considered the therapy of choice for this syndrome, a subset of patients may respond to interleukin-6 antagonists instead. However, this approach does not reduce M-proteins and has no impact on the natural history of the disease. An international registry for patients with Schnitzler’s syndrome is ongoing, joining the different specialties involved in the care of these patients. The aim of the registry is to facilitate the collection of standardized information, enabling collaborative research and helping unveil new evidence in this complex autoinflammatory disorder [75].

## 7. Monoclonal Gammopathies of Ocular Significance

The cornea is normally a transparent structure. Several abnormalities can cause corneal opacities, making vision difficult. Patients with MG should be included in the differential diagnosis of acquired corneal opacities, as this ocular finding could be the initial manifestation of a systemic disease that can potentially be life threatening. As happens in other groups of MGCS with kidney or skin biopsies, corneal biopsy is of great diagnostic value. However, when it is not feasible due to the location of the corneal pathology, aqueous sampling may be an alternative approach for diagnostic purposes. The term MGOS was proposed for patients diagnosed with MGUS, in which the only significant clinical finding is ocular manifestation [54,76,77,78]. A regular, yearly ophthalmic checkup of these patients to improve their quality of life has been suggested.

MGOS is a rare subset of MGCS that occurs secondary to PC disorders and causes ocular manifestations. The most frequent ocular M-protein–related condition is crystalline keratopathy consisting of Ig deposition, corneal thickening, photophobia, and finally, visual loss. In a recent series of 23 patients with paraproteinemic keratopathy, neither ocular nor hematologic treatment afforded a durable improvement in visual acuity (recurrence after a median of 11 months), despite initial responses. Further studies are required to determine the optimal strategy to treat and prevent the relapse of ocular symptoms in patients with paraproteinemic keratopathy [79].

## 8. Monoclonal Gammopathies of Neurological Significance

Peripheral neuropathy (PN) is commonly encountered in clinical practice. It is defined as a disease or degenerative state of the peripheral nerves in which motor, sensory, or vasomotor nerve fibers are affected. Patients can refer to muscle weakness, pain, and numbness [80,81]. Several MGs are associated with PN, and in this context, the constellation of neurological symptoms is often referred to as paraproteinemic neuropathy. Neuropathy as end-organ damage in MGs has received relatively little attention, partly due to the risk of performing nerve biopsy. Neuropathies at times tend to be severe, even though hematological involvement remains latent. Since most patients may have a severe disability, an argument in favor of a new classification of these entities, such as MGNS, was suggested. Every approach in this setting should focus on early diagnosis in order to avoid the development of severe nerve damage and its consequent impact on quality of life. As occurs in other groups of MGCS, a multidisciplinary approach between hematologists and neurologists is crucial to correctly diagnose and treat MGNS.

Almost 10% of patients with neuropathy of unknown cause have an M protein. It is well known that the presence of PN is particularly frequent in patients with IgM MGs. The first step is to exclude common causes of secondary PN. The exact prevalence of MGNS is unknown and likely underrated [82,83,84,85]. PN is not associated with an increased risk of progression to MM, but a 2.9-fold risk of AL amyloidosis was found in a large population-based study [83].

The efficient management of IgM patients with PN confirmed by neurophysiological tests depends on the type of neuropathy: Demyelinating cases with prolonged distal latency must be tested for anti-myelin-associated glycoprotein antibodies to rule out anti-myelin-associated glycoprotein neuropathy; negative cases with chronic ataxia and ophthalmoplegia must be tested for antiganglioside antibodies to exclude CANOMAD (chronic ataxic neuropathy, ophthalmoplegia, Ig M paraprotein, cold agglutinins, and disialosyl antibodies). In double-negative patients, chronic inflammatory demyelinating polyradiculoneuropathy or paraproteinemic PN are the most plausible scenarios. -On the contrary, in patients with axonal or mixed (both axonal and demyelinating characteristics) NP, cryoglobulinemia, and amyloidosis should be considered. Finally, a nerve biopsy should be reserved for atypical or very aggressive cases with an elusive diagnosis. The accurate diagnosis of MGCS commonly depends on demonstrating specific organ damage through a biopsy. Skin and kidney biopsies are relatively safe procedures. However, a nerve biopsy can be associated with permanent disability in terms of sensory or motor deficits and pain. Therefore, nerve biopsy is less desirable as a diagnostic method, and the decision to perform it must be made after a detailed risk and benefit analysis. Without the routine use of definitive methods to establish a relationship between the M-protein and PN, the diagnosis of MGNS would be one of exclusion.

CANOMAD is a rare syndrome characterized by chronic neuropathy with sensory ataxia, ocular, and/or bulbar motor weakness in the presence of a monoclonal IgM reacting against gangliosides containing disialosyl epitopes [86]. In a French multicenter retrospective study that included 45 patients, 91% required treatment. Intravenous immunoglobulins and rituximab-based regimens were the most effective therapies.

## 9. MGCS as a Global and Unifying Concept

### 9.1. Diagnosis

The diagnosis of MGCS is a complex process. The common clinical challenge is trying to demonstrate a causal link between specific organ damage and the presence of an M-protein. Performing a tissue or organ biopsy is needed in most cases in order to achieve this purpose, nowadays still using conventional immunohistological techniques. The current pathophysiological classification of MGCS [50] is shown in Table 8.

A high titer of autoantibody activity is important for the diagnosis of an M-Ig-mediated immune process. Moreover, the specificity of M-Ig should be defined and correlated with clinical data. Other complementary immunological tests can be of help in the final diagnosis of some specific entities, such as complement studies in type II cryoglobulinemia and xanthomatosis. The pathophysiological mechanism of many MGCS still remains unknown. Epidemiology plays an important role in the definition of the entity in these cases. For certain entities, clinical-based evidence is the main available proof supporting the link between organ damage and the M-protein. This is the case with TEMPI syndrome (Telangiectasias, elevated erythropoietin level and Erythrocytosis, Monoclonal gammopathy, Perinephric fluid collections, and Intrapulmonary shunting). The first patient with TEMPI syndrome was described in 2010 [87]. It is an ultrarare novel multisystem disease [88,89,90,91]. Only 22 patients were reported at the end of 2019. The current proposed diagnostic criteria are outlined in Table 9.

The diagnosis of TEMPI syndrome is challenging in many ways, mainly due to its rarity and the dynamics of diagnostic criteria. The average diagnostic delay of the reported cases is ten years. The presence of MG plays a central role in this syndrome, besides erythrocytosis and telangiectasias, which are commonly present from the beginning. The erythropoietin level tends to progressively increase, preceding the development of the features of intrapulmonary shunting and perinephric fluid collections. Despite the proposed set of diagnostic criteria, no specific criteria for the definition have been outlined to date. Bone marrow biopsy typically shows findings consistent with those of MGUS or smoldering myeloma [92]. Initial treatment with daratumumab or bortezomib is recommended, but other drugs and autologous stem cell transplants are also effective [93,94,95,96,97]. The increase of serum erythropoietin or the M-protein, as well as the reemergence of telangiectasias, may be used as markers of relapse. Knowledge about TEMPI syndrome is in continuous evolution [98,99,100,101,102]. Participation in the international registry is encouraged.

The MGCS concept represents an important step forward in the field of MGs, increasing clinicians’ awareness of the fact that the size of the M-protein is not always related to the clinical impact. A timely strategy to confirm a link between the M-protein and system, tissue, or organ damage should be established as soon as possible in order to avoid further impairment. Clinical manifestations may overlap with other unrelated M-protein diseases, making diagnosis and clinical decisions difficult [103].

The subset of MGCS sharing the pathophysiological mechanism of autoantibody activity deserves special attention. This group includes a wide array of entities, from specific hematological disorders such as hemolysis or bleeding to the involvement of only one organ or the presence of systemic disease. M-protein-associated bleeding disorders have been essentially reported with auto-antibodies directed against von Willebrand factor or factor VIII, but an auto-anti-thrombin antibody has also been described recently for the first time [104].

### 9.2. Treatment

Treatment of MGCS focuses on the presence of symptoms and disability. Therapy should be based first on a risk-to-benefit assessment. The next step should be the consideration of the isotype of the M-protein involved, provided that non-Ig-M entities are usually managed with anti-MM therapy, whereas Ig-M ones are mainly treated with an anti-CD20 approach. The primary objective is achieving a hematological response, which is necessary to achieve later an organ response. Future directions need to be focused on the pathophysiology of the disease, exploring new therapeutic options through a deep understanding of immune background dysregulation, and trying to predict which patients might develop MGCS. Specific clinical or laboratory features to identify which patients are at higher risk of developing MGCS are lacking. On the other hand, no standardized scores to assess the risk of progression have been reported to date. Few reports have attempted to describe the risk of progression from MGCS to symptomatic MM or other lymphoproliferative disorders [105].

There are lights and shadows in the MGCS setting. Several ongoing large-scale prospective studies will help to reveal important knowledge gaps. The Iceland Screens Treats and Prevents Multiple Myeloma (iSTOPMM; NCT03327597) study [106] screens individuals who are at least 45 years old for MGUS, followed by the randomization and evaluation of intensive follow-up versus standard follow-up. The Predicting Progression of Developing Myeloma in a High-Risk Screened Population (PROMISE; NCT03689595) study [107] aimed to assess the prevalence of MGUS in a population at high risk of MM, confirming a high prevalence in older adults who were black/African Americans or who had some first-degree relative with hematologic malignancy. This population may benefit from specific screening, allowing early treatment. Finally, the main objective of the NoMoreMGUS study in Spain [108] is to develop minimally invasive methods for the differential diagnosis between MGs with and without clinical significance. This will be accomplished by integrating clinical data with new minimally invasive biomarkers for prospective identification of individual MGUS cases at ultra-high risk of transformation or life-threatening infection, offering precision medicine to prevent disease progression or complications, and increasing cure rates. The study also aims to generate longitudinal tumor and immune data for the identification of specific targets towards individualized therapies aimed at the prevention of malignant cancer or life-threatening infection.

The current place of MGCS in the global scenario of MGs is represented in Figure 2.

## 10. Discussion and Practical Applications

MGs represent a broad differential diagnosis in daily clinical practice, requiring a thorough evaluation for “clinical significance”. MGCSs are a heterogeneous collection of old and new diseases associated with a nonmalignant clone composed of B cells or cBMPCs that produce M-proteins, and finally, the establishment of selective or systemic organ damage through diverse and sometimes poorly understood mechanisms [109]. The concept of MGCS was introduced as an extension of the already established concept of MGRS. Because of its therapeutic implications, MGCS must be considered a separate entity within the spectrum of MGs [50]. Some controversial aspects are highlighted below.

### 10.1. Classification

The concept of MGCS was coined in the last decade. The right place of MGCS in the current and future International Classification of Diseases remains to be defined. New conditions included in the updated classification of PCNs of WHO-HAEM5 are MGRS and AESOP (adenopathy and extensive skin patch overlying a plasmacytoma) syndrome [1,110]. However, the Clinical Advisory Committee of the International Consensus Classification of Mature B-cell Neoplasms [2] states that both MGRS and MGCS do not represent separate disease entities, but are descriptive terms that can be added as clinical features to the underlying diagnosis (e.g., MGUS). Therefore, the integration of both international classifications is complex and to some extent contradictory. From a clinical point of view, growing evidence supports considering MGCS as a different entity, as it was proposed [50], based on the poor outcome compared with MGUS and the need for therapy. However, controversy remains.

### 10.2. Definition

Heterogeneity is a hallmark of MGCS that encompasses a large group of entities with different epidemiological backgrounds, clinical presentations, pathogenetic mechanisms, and prognostic impacts. Therefore, the definition must be adapted to this wide scenario. Overall, MGCS is a global term used to describe patients with nonmalignant MGs with unexpected important diseases. MGCS should be excluded in any patient presenting with a possible MGUS who is also experiencing other unexplained symptoms. The first step in making these diagnoses is to consider them. Several entities among the MGCSs are complex syndromes; therefore, recognizing major but also other minor manifestations is crucial for achieving a timely diagnosis [51]. Proving a causal relationship between MG and clinical manifestations is not an easy process and finally may depend on a biopsy, showing the pathological M-Ig deposition in the involved tissue or organ [50].

### 10.3. Epidemiology

Our current knowledge about the clinical and genomic epidemiology of MGCS is scarce. The true incidence and prevalence of most MGCSs remain largely unknown. This is due to several factors. First, most MGCS cases derive from the study of MGUS, and this entity is not usually covered in population-based registries. Second, MGRS and MGSS depend on the performance of a biopsy, and the likelihood of undergoing a kidney biopsy varies [48,57,58]. Even in patients with MG, the rate of performing a kidney biopsy is quite low, at 2.5%; this rate increases to 10% in the chronic kidney disease population [58]. Therefore, some MGCSs may be underreported. Third, most of the published data are derived from case reports and single-center series. Therefore, caution should be taken for suggested recommendations. Fourth, clinical trials that focus on MGCS are almost anecdotal.

AL amyloidosis plays a central role in patients with MGCS, particularly MGRS [57,58,68]. The epidemiology of AL has just been reviewed [111]. The epidemiological estimates for AL amyloidosis in this study highlight its global low incidence, helping to determine the true burden of the disease, which is key for adapting research and health care resources.

### 10.4. Clinical Presentation and Final Diagnosis

Several MGCSs are syndromes. These disorders are usually separated into symptoms and signs referable to the nerves, kidneys, skin, or eyes, but recognizing other manifestations is also key [51]. The impact of frailty and comorbidities must be taken into account in both MGUS and MGCS [112], particularly in the elderly population. Hematologists should work together with colleagues from other specialties (Internal Medicine, Nephrology, Neurology, Rheumatology, Dermatology, Ophthalmology, Pathology, and others) in order to offer an optimized approach to MGCS patients.

The risk of bacterial and viral infections in MGUS is increased. However, a recent study showed that MGUS does not affect susceptibility to SARS-CoV-2 or the severity of COVID-19 [113]. Ongoing studies will determine whether a subgroup of MGUS or MGCS patients may be characterized by a higher infectious risk.

On the other hand, some recent reports emphasize the likelihood of finding MGCS behind concrete rare clinical scenarios, such as the case of unexplained myopathy [114].

Rheumatologists play an important role in the diagnosis and management of a group of MGCS who may present with a systemic inflammatory picture, such as scleromyxedema, Schnitzler’s syndrome, idiopathic systemic capillary leak syndrome (Clarkson’s disease), and TEMPI syndrome [115].

Overall, the main objective is to obtain a correct and timely diagnosis. This may be achieved through a comprehensive and risk-adapted workup. The use of new technologies is encouraged. Until a minimally invasive diagnostic approach using peripheral blood can be confirmed as a validated alternative, bone marrow examination is still necessary to confirm the established diagnostic criteria and obtain complementary diagnostic and prognostic tests.

### 10.5. Dynamics of Diagnostic Criteria

Many MGCS conditions have a clinical presentation as a set of signs and symptoms that occur together and usually form an identifiable pattern. This syndrome characterizes a specific condition. However, not all signs or symptoms are present at the same time at baseline. For instance, the typical manifestations of TEMPI syndrome include the pentad of telangiectasias, elevated erythropoietin and erythrocytosis, monoclonal gammopathy, perinephric fluid collections, and intrapulmonary shunting. However, not all features are commonly present at the onset of the disease [91]. Moreover, other important features not included in the TEMPI acronym, such as venous thrombosis, spontaneous intracranial hemorrhage, ascites, and other cutaneous [99] or ocular signs [101], may also appear. Therefore, there is a need to face the clinical presentation of patients dynamically and individually rather than trust the diagnosis only in a set of diagnostic criteria that do not have to be completely fulfilled in a specific moment of the evolution of the disease. This approach will probably avoid misdiagnosis and diagnostic delays.

### 10.6. Risk of Progression

The information about the risk of progression in MGCS is limited. For MGRS, this risk seems to be higher than for MGUS [61]. Adequate follow-up in people with MGUS has shown a clinical benefit in terms of less serious complications and longer survival [116]. Whether a risk-adapted follow-up of MGCS patients can translate into better survival remains to be determined. The ongoing iSTOPMM study [106] will give insight into this important topic.

A shift to a minimally invasive approach to the risk of progression using peripheral blood instead of BM is expected to occur in the coming years. Circulating tumor cells (CTCs), as defined by the presence in peripheral blood of cPCs, are a powerful prognostic marker in MM [117]. The frequency and number of CTCs using next-generation flow cytometry were investigated for the first time in PCNs. CTCs were detected in 100% of MM, 100% of SMM, and 59% of MGUS. Higher levels of CTCs were associated with higher levels of BM infiltration, more adverse prognostic features, and a shorter time to progression from MGUS to MM. Therefore, growing evidence supports the key prognostic role of baseline CTC evaluation by next-generation flow in the full range of PCN [118,119,120]. Molecular hallmarks of CTCs were investigated, showing that CTCs overexpressed genes involved in inflammation, hypoxia, or epithelial–mesenchymal transition, whereas genes related to proliferation were downregulated [121]. In fact, the evaluation of CTCs in peripheral blood outperformed the quantification of cBMPCs. A recent study showed that the presence of ≥0.01% CTCs in transplant-eligible MM patients could be a new risk factor in novel staging systems [122].

### 10.7. Laboratory Biomarkers

The clinical laboratory plays a critical role in the diagnosis, monitoring, prognosis, and response evaluation of MGs. More sensitive tests, such as mass spectrometry, in addition to the analysis of CTCs, immune profiling, a multi-omics approach, and the clinical use of new biomarkers, will help to enhance our current knowledge of these disorders. The potential impact of CTCs in peripheral blood by next-generation flow in both MGUS and MGCS will be evaluated in an ongoing large-scale NoMoreMGUS study [108]. On the other hand, the use of quantitative matrix-assisted laser desorption ionization-time of flight (MALDI-TOF) mass spectrometry (lower limit of M protein quantification 0.015 g/L) in the PROMISE study allowed the description of the called MG of indetermined potential, including cases of MG detected and quantified at concentrations of 0.015–0.2 g/L. It remains unknown whether this disorder is an earlier precursor state along the MGUS to MM continuum [107].

Distinct biomarker trajectories in patients with MGUS over time have been identified [123]. Future research should investigate how these trajectories may be related to the risk of MM progression.

Remarkably, a recent study using Single-Molecule Real-Time Sequencing gives a step forward in the application of personalized medicine to the MGs field, showing that the M-protein of each patient behaves as a tumoral fingerprint, allowing the traceability of patients’ specific M-proteins at diagnosis and after therapy [124].

### 10.8. The Importance of National and International Registries

The use of prospective clinical-epidemiological registries is a fundamental tool to advance the knowledge of diseases with a low incidence. Scandinavian MG registries illustrate the benefits offered by this strategy in the last decades. Unfortunately, MGUS cases are not included in the population-based registries of most countries around the world. Therefore, large-scale projects such as the iSTOPMM, PROMISE, and NoMoreMGUS studies are an excellent opportunity to unveil new data and insight on this topic. In addition to these projects, national and international monographic registries should be used to deepen the knowledge of ultra-low incidence disorders, as is the case for CANOMAD or TEMPI syndromes. In addition, specific MGCS clinical trials are encouraged.

### 10.9. IgM MGCS Idiosyncrasy

IgM MGUS comprises around 15–20% of all cases of MGUS. It is distinct from other forms of MGUS in that the typical primary progression events include mainly Waldenström macroglobulinemia and AL amyloidosis. In addition, IgM MGUS is more commonly associated with autoimmune phenomena, resulting in distinctive clinical pictures. IgM-related MGCS include some peculiar entities such as cold agglutinin disease, IgM-related neuropathies, renal manifestations, and Schnitzler’s syndrome. The diagnostic approach to, and management of this subgroup of disorders may differ from other categories of MGCS. In particular, the treatment for IgM MGCS should be adapted based on the organ(s) involved, symptoms, and patient fitness [125].

A significant proportion of IgM MGUS develop unique immunological and biochemical manifestations, which are M-protein related, in the absence of overt malignancy and are termed IgM-related MGCS. Five important subtypes have just been reviewed: cold agglutinin disease, type I and II cryoglobulinemia, IgM-associated PN, Schnitzler syndrome, and IgM-associated AL amyloidosis [126]. There is an IgM kappa predominance in all cases, except IgM-associated AL amyloidosis, which is increasingly recognized as a distinctive entity [127,128].

Assessment of *MYD88*^L265P^ has been shown to improve lymphoma diagnosis and treatment and is becoming increasingly requested for certain entities, such as IgM monoclonal disorders [39].

### 10.10. An Evidence-Based Approach to Standardize Therapy

Unlike people with MGUS, MGCS patients require therapy. However, the treatment of most of these disorders is far from being standardized due to its heterogeneity and rarity. Despite the absence of solid evidence to make clinical decisions in most scenarios, the treatment possibilities for MGCS have recently been summarized [105]. The transfer of the patient to an experienced center offering multidisciplinary support or expert advice is recommended. Collaborative research is imperative to aid in defining optimal treatment strategies. Early diagnosis is key to offering timely treatment. Therefore, the judicious use of conventional diagnostic methods, such as screening in selective clinical scenarios, could help to avoid diagnostic delay [129].

## 11. Conclusions

The field of MGs is continuously evolving. MGs range from transient MGUS to plasma cell leukemia. MGUS is the most frequent MG, with an increasing prevalence with aging. Despite being considered a pre-malignant asymptomatic disorder, people with MGUS have decreased life expectancy due to malignant transformation and non-malignant causes. MGUS is a large and heterogeneous group. MGCS is a young concept encompassing a constellation of diseases associated with a nonmalignant B cells/PCs clone that produces M-protein and a pathology through diverse, ill-defined mechanisms. Therefore, MGCS patients are characterized by the presence of a quiescent and generally small B-cell/PC non-malignant clone and symptoms that are related to the M-protein or to the clone itself by mechanisms other than the tumor burden. A kidney or skin biopsy is needed for MGRS and MGSS, respectively. Other tissue biopsies could eventually be necessary.

Our current knowledge of the epidemiology and pathogenesis of MGCS is limited due to its heterogeneity and relatively low frequency. There are many syndromes among the spectrum of MGCS. The diagnosis of these disorders is based on a set of criteria that are not always present at baseline, making the diagnosis process difficult. New technologies, including mass spectrometry, multi-omics, and a shift to a minimally invasive approach for diagnosis, will help in making an easier and efficient diagnosis. Ongoing large-scale studies will unveil new insights on the topic in the coming years. In the meantime, a multidisciplinary and collaborative approach is mandatory, participating in national and international monographic registries, as well as in specific clinical trials. Only through a comprehensive and integral clinical, biological, and epidemiological study of each patient can we generate knowledge to offer the best evidence-based treatment to the right patient at the right time.

## Figures and Tables

**Figure 1 cancers-14-05247-f001:**
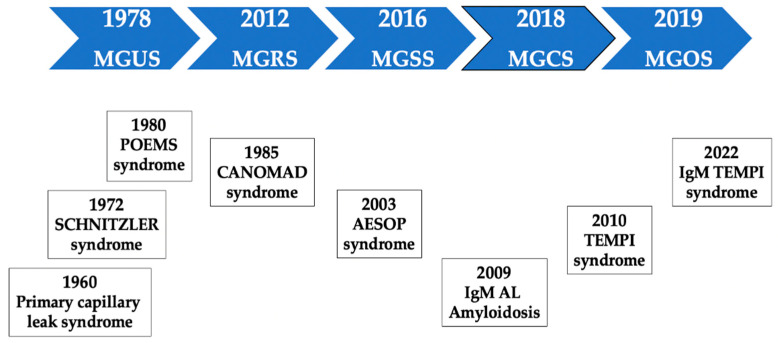
Schematic timeline of selected MGCS disorders. Abbreviations: AESOP; adenopathy and extensive skin patch overlying a plasmacytoma, CANOMAD; chronic ataxic neuropathy, ophthalmoplegia, monoclonal IgM protein, cold agglutinins, anti-disialosyl antibodies MGs; monoclonal gammopathies, MGCS; MG of clinical significance, MGOS; MG of ocular significance, MGNS; MG of neurological significance ^†^, MGRS; MG of renal significance, MGSS; MG of skin significance, MGUS; MG of undetermined significance, POEMS; polyradiculoneuropathy (P), organomegaly (O), endocrinopathy(E), monoclonal gammopathy (M), and skin changes (S) syndrome, TEMPI; Telangiectasias, elevated erythropoietin level and Erythrocytosis, Monoclonal gammopathy, Perinephric fluid collections, and Intrapulmonary shunting. ^†^ MGNS is not a term formally coined in a specific moment of time. Before the term MGCS was coined, the name “paraproteinemic neuropathy” was commonly used to describe the association (not always causal) between MG and neuropathy. Since 2018, MGNS has been an increasingly used term, and some old entities, such as CANOMAD syndrome, have been recognized as belonging to this group.

**Figure 2 cancers-14-05247-f002:**
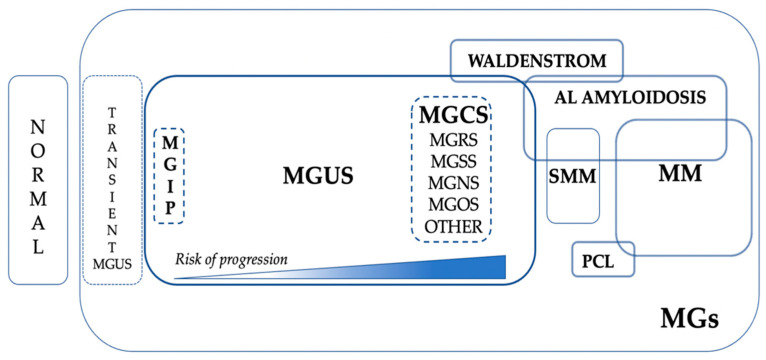
The global scenario of monoclonal gammopathies. Abbreviations: MGs; monoclonal gammopathies, MGCS; MG of clinical significance, MGIP; MG of indetermined potential, MGOS; MG of ocular significance, MGNS; MG of neurological significance, MGRS; MG of renal significance, MGSS; MG of skin significance, MGUS; MG of undetermined significance, MM; multiple myeloma, PCL; plasma cell leukemia, SMM; smoldering MM.

**Table 1 cancers-14-05247-t001:** Plasma cell neoplasms and other diseases with paraproteins.

Monoclonal Gammopathies
▪ Cold agglutinin disease *
▪ IgM monoclonal gammopathy of undetermined significance
▪ Non-IgM monoclonal gammopathy of undetermined significance
▪ Monoclonal gammopathy of renal significance *
**Diseases with monoclonal immunoglobulin deposition**
▪ Immunoglobulin-related (AL) amyloidosis
▪ Monoclonal immunoglobulin deposition disease
**Heavy chain diseases**
▪ Mu heavy chain disease
▪ Gamma heavy chain disease
▪ Alpha heavy chain disease
**Plasma cell neoplasms**
▪ Plasmacytoma
▪ Plasma cell myeloma
▪ Plasma cell neoplasms with associated paraneoplastic syndrome
◊ POEMS
◊ TEMPI
◊ AESOP *

Adapted from Alaggio et al. [1]. * Not previously included. POEMS: polyneuropathy, organomegaly, endocrinopathy, monoclonal gammopathy, and skin abnormalities; TEMPI: telangiectasias, elevated erythropoietin and erythrocytosis, monoclonal gammopathy, perinephric fluid collection, and intrapulmonary shunting; AESOP: adenopathy and extensive skin patch overlying a plasmacytoma.

**Table 2 cancers-14-05247-t002:** International Consensus Classification of Mature B-cell Neoplasms.

IgM Monoclonal Gammopathy of Undetermined Significance (MGUS)
▪ IgM MGUS, plasma cell type *
▪ IgM MGUS, NOS *
**Primary cold agglutinin disease ***
**Heavy chain diseases**
▪ Mu heavy chain disease
▪ Gamma heavy chain disease
▪ Alpha heavy chain disease
**Plasma cell neoplasms**
▪ Non-IgM monoclonal gammopathy of undetermined significance
▪ Multiple myeloma (Plasma cell myeloma) *
◊ Multiple myeloma NOS
◊ Multiple myeloma with recurrent genetic abnormality
- Multiple myeloma with CCND family translocation
- Multiple myeloma with MAF family translocation
- Multiple myeloma with NSD2 translocation
- Multiple myeloma with hyperdiploidy
▪ Solitary plasmacytoma of bone
▪ Extraosseous plasmacytoma
**Monoclonal immunoglobulin deposition diseases**
▪ Immunoglobulin light chain amyloidosis (AL) *
▪ Localized AL amyloidosis *
▪ Light chain and heavy chain deposition disease

Adapted from Campo, E. et al. [2]. * Changes from the 2016 WHO classification; NOS: not otherwise specified; CCND: cyclin D; MAF: musculoaponeurotic fibrosarcoma; NSD2: nuclear receptor SET domain-containing.

**Table 3 cancers-14-05247-t003:** Variables influencing the risk of progression in individuals with MGUS.

Variable	References
Serum M protein size (≥15 g/L)	[11,12,18]
Serum M protein type (non-IgG MGUS)	[11]
Abnormal serum FLCr	[11,12]
Immunoparesis (classical)	[12,15]
% immunophenotypically aberrant BMPC (<95% vs. ≥95%)	[13,14]
DNA index (diploid vs. aneuploid)	[13]
Evolving type	[14]
Immunoparesis (HLC)	[16]
iFLC (>100 mg/L)	[18]
Chromosomal abnormalities on purified BMPC (FISH)	[19,22,23]
Serum BCMA level	[20]
Copy number variants (1q21 gain and 13q deletion) on CD138 + BMPC (digital PCR)	[21]
Circulating Tumor Cells	[24]
Body mass index	[25,26]

BCMA: B-cell maturation antigen; BMPC: bone marrow plasma cells; FISH: fluorescence in situ hybridization; FLCr: free light chain ratio; HLC: heavy light chain; iFLC: involved free light chain; PCR: polymerase chain reaction.

**Table 4 cancers-14-05247-t004:** Classification of MGRS-associated diseases.

M Ig Deposits				No M Ig Deposits
Organized			Non-Organized	
Fibrillar	Microtubular	Inclusions or crystalline deposits	MIDD	C3 glomerulopathy with MG
Ig-related amyloidosis	Immunotactoid GN	LCPT	PGNMID	Thrombotic microangiopathy
M fibrillary GN	Cryoglobulinaemic GN	Crystal storing histiocytosis	Miscellaneous	
		Cryo crystalglobulin GN		

Adapted from Leung, N. et al. [48]. GN: glomerulonephritis; Ig: immunoglobulin; LCTP: light-chain proximal tubulopathy; M: monoclonal; MG: monoclonal gammopathy; MIDD: monoclonal immunoglobulin deposition disease; PGNMID: proliferative glomerulonephritis with monoclonal immunoglobulin deposits.

**Table 5 cancers-14-05247-t005:** Classification of MGSS-associated diseases.

Group	Association	Disorders
**I**	Malignant PCN with cutaneous manifestations	Waldenström macroglobulinemia
		AL Amyloidosis
		Cryoglobulinemia
		Plasmacytoma
		POEMS syndrome
**II**	Non-malignant MG High association	Scleromyxedema
		Scleredema
		Necrobiotic xanthogranuloma
		Plane xanthoma
		Schnitzler syndrome
**II**	Non-malignant MG Low association	Pyoderma gangrenosum
		Sweet syndrome
		Leukocytoclastic vasculitis
		Neutrophilic dermatosis
**II**	Non-malignant MG Unknown	Erythema elevatum diutinum
		Subcorneal pustular dermatosis
**III**	Anecdotal	Miscellaneous cutaneous disorders described in association with MGs
**IV**	Not specific for MG	Miscellaneous cutaneous signs or symptoms including purpura, pruritis, infection, adverse reactions to medications, etc.

Adapted from Daoud et al. [69]. MG: monoclonal gammopathy; PCN: plasma cell neoplasms; POEMS: polyradiculoneuropathy (P), organomegaly (O), endocrinopathy (E), monoclonal gammopathy (M), and skin change (S) syndrome.

**Table 6 cancers-14-05247-t006:** POEMS syndrome diagnostic criteria.

Major-Mandatory (Both Required)	Polyneuropathy
	Monoclonal Plasma Cell-Proliferation Disorder
Major-Other (at least one required)	Castleman’s disease
	Sclerotic bone lesions
	Vascular endothelial growth factor elevation
Minor (at least one required)	Organomegaly (splenomegaly, hepatomegaly, or lymphadenopathy)
	Extravascular volume overload (edema, pleural effusion, or ascites)
	Endocrinopathy (adrenal, thyroid, pituitary, gonadal, parathyroid)
	Skin changes
	Papilledema
	Thrombocytosis
Other symptoms and signs	Clubbing, weight loss, pulmonary hypertension, restrictive pulmonary syndrome, diarrhea, thrombotic disease, hyperhidrosis

Adapted from Dispenzieri et al. [71].

**Table 7 cancers-14-05247-t007:** Schnitzler syndrome diagnostic criteria.

Major-Mandatory	Chronic Urticarial Rash
	IgM or IgG Monoclonal Gammopathy
Minor	Recurrent fever
	Abnormal bone remodeling with or without pain
	Neutrophilic dermatosis on skin biopsy
	Leukocytosis and/or elevated C-reactive protein

Adapted from Simon et al. [74].

**Table 8 cancers-14-05247-t008:** Pathophysiological classification of MGCS.

Subtype	Mechanism	Main Organ Involved
AUTOANTIBODY		
Type II Cryoglobulinemia	Rheumatoid	S, K, PN, SYS
C1 inhibitor deficiency	C1 inhibitor	Angioedema
vWD	vWF	Bleeding
Bullous diseases	Collagen VII	S
Xanthomatosis	Lipoproteins	S, tendons, other
Cold agglutinin disease	RBC	S, hemolysis
IgM-associated PN	MAG, gangliosides	PN, CANOMAD
CAP ACTIVATION		
C3 glomerulonephritis		K, SYS
Atypical HUS		
CYTOKINE MEDIATED		
POEMS syndrome	VEGF	POEMS syndrome
UNKNOWN		
SCL syndrome		SYS
TEMPI syndrome		SYS
Neutrophilic dermatosis		S
Acquired cutis laxa		S
Scleromyxedema		S
Scleroedema		S
Schitzler syndrome	IL-1 deregulation?	S, SYS
Nemaline myopathy		M

Adapted from Fermand, J. P. et al. [50]. CAP, complement alternative pathway; CANOMAD, chronic ataxic neuropathy, ophthalmoplegia, monoclonal IgM protein, cold agglutinins, anti-disialosyl antibodies; HUS, hemolytic-uremic syndrome; IL-1, interleukin 1; K, kidney; M, muscle; MAG, myelin-associated-glycoprotein; PN, peripheral neuropathy; POEMS, polyneuropathy, organomegaly, endocrinopathy, monoclonal gammopathy, and skin changes; RBC, red blood cell; S, skin; SCL, systemic capillary leak syndrome; SYS, systemic; TEMPI, telangiectasias, erythrocytosis with elevated erythropoietin level, monoclonal gammopathy, perinephric fluid collection, and intrapulmonary shunting; VEGF, vascular endothelial growth factor; vWD, von Willebrand disease; vWF, von Willebrand factor.

**Table 9 cancers-14-05247-t009:** TEMPI syndrome diagnostic criteria.

Major	Telangiectasias
	Elevated erythropoietin and erythrocytosis
	Monoclonal gammopathy
Minor	Perinephric fluid
	Intrapulmonary shunting
Other	Venous thrombosis

Adapted from Sykes et al. [89].

## Data Availability

Not applicable.

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
