# Peer review of "Monoclonal Gammopathies of Clinical Significance: A Critical Appraisal"

_cancers, 2022, doi:10.3390/cancers14215247_

Round 1

Reviewer 1 Report

This is a brilliant review with a really high clinical significance.

Just a little bit of notes:

11)      In figures 1 and 2 some of words are underlined with red curly curve for some reason.

22)      There is an abbreviation MGNS in the caption to Figure 1 but it missed in this Figure.

33)      Please add abbreviations MAG and VEGF to notes in Table 8.

44)      There is a misprint “iincreasing” in line 426.

Reviewer 2 Report

This is a timely review of a complex topic. The Authors have provided a comprehensive and erudite manuscript which will be of interest to a range of clinicians and scientists working in the field. They correctly emphasise the need for a multidisciplinary approach. The text is well written, with good use of Tables. 

Minor points:

1. There are occasional very minor typographical/grammatical errors. For example, line 191 should read 'understood or unknown', rather than 'understood of unknown', and on line  275 'one major criteria' and one 'minor criteria' should read 'one major criterion' and 'one minor criterion'. I assume examples such as these would be picked up at proof reading stage, if accepted for publication.

2. As a personal opinion, I question the use of 'non-malignant' when describing disorders which have the potential to be life threatening.

3. The potential benefits of including serum protein electrophoresis (and possibly free light chain assays) in blood profiles performed during emergency and elective admissions to hospital should be highlighted. See: Atkin C et al. The prevalence and significance of monoclonal gammopathy of undetermined significance in acute medical admissions. Br J Haematology 2020;189,6:1127-1135. Especially pertinent in context of admissions with chronic kidney disease and heart failure, where the underlying aetiology may be a monoclonal gammopathy which otherwise may go undetected if not screened for.

Overall then, the Authors have done a fine job of presenting a complex topic, well referenced and in readable form.

Reviewer 3 Report

This is a very well writen and intensive review of MGUS and MGCS. I have only few suggestions.

1. In the Figure 1: it has abbreviation of "MGNS; MG of neurological significance" but it was not contain in the figure

2. Page 4 line 95 - 97: This is a very important point to emphasize that although low-risk MGUS, there was a study demonstrated as high as half of patient had BM plasma cell reach the criteria of SMM.

3. Table3: serum FLCr -> abnormal SFLCr

4. Table 4: MGRS: Please add the line to separate between colume of organize and non-organize

5. Page 13 and line 426: iincrease -> increase

6. The controvery section

- The diagnosis: page 15 line 538 - 541: it is about the increase risk of complications of MGUS. This may be more suitable to separate the topic: such as MGUS increase risk of thrombosis, fracture, and infection..etc.
